# Potential of Polyphenols to Restore SIRT1 and NAD+ Metabolism in Renal Disease

**DOI:** 10.3390/nu14030653

**Published:** 2022-02-03

**Authors:** Claudia Tovar-Palacio, Lilia G. Noriega, Adriana Mercado

**Affiliations:** 1Dirección de Nutrición, Instituto Nacional de Ciencias Médicas y Nutrición Salvador Zubirán, Mexico City 14080, Mexico; 2Departamento de Fisiología de la Nutrición, Instituto Nacional de Ciencias Médicas y Nutrición Salvador Zubirán, Mexico City 14080, Mexico; lilia.noriegal@incmnsz.mx; 3Laboratorio de Biología Molecular, Departamento de Nefrología, Instituto Nacional de Cardiología Ignacio Chávez, Mexico City 14080, Mexico

**Keywords:** Sirt1, NAD^+^, AKI, obesity, DKD, polyphenols

## Abstract

SIRT1 is an NAD^+^-dependent class III histone deacetylase that is abundantly expressed in the kidney, where it modulates gene expression, apoptosis, energy homeostasis, autophagy, acute stress responses, and mitochondrial biogenesis. Alterations in SIRT1 activity and NAD^+^ metabolism are frequently observed in acute and chronic kidney diseases of diverse origins, including obesity and diabetes. Nevertheless, in vitro and in vivo studies and clinical trials with humans show that the SIRT1-activating compounds derived from natural sources, such as polyphenols found in fruits, vegetables, and plants, including resveratrol, quercetin, and isoflavones, can prevent disease and be part of treatments for a wide variety of diseases. Here, we summarize the roles of SIRT1 and NAD^+^ metabolism in renal pathophysiology and provide an overview of polyphenols that have the potential to restore SIRT1 and NAD^+^ metabolism in renal diseases.

## 1. Introduction

Sirtuins constitute a family of nicotinamide adenine dinucleotide (NAD^+^)-dependent deacetylases that have been conserved during evolution from archaea to mammals; their activity is regulated by metabolic and environmental stimuli [1]. Mammals carry seven sirtuins (SIRT1–7) with diverse physiological functions and cellular locations [2,3]. In particular, sirtuin 1 (SIRT1), which is localized to both the nucleus and the cytoplasm [4,5], is expressed in embryonic and adult tissues such as the brain, pancreas, liver, and kidney [6,7,8]. In all these tissues, SIRT1 has been linked to gene transcription, cell senescence, energy homeostasis, glucose metabolism, and insulin secretion [9,10]. Several reports have described the roles of sirtuins in renal pathophysiology, and SIRT1 has been the most extensively studied sirtuin in the kidney (see reviews by [11,12,13]).

The kidney is one of the most energy-demanding organs in the human body due to the high-energy requirements for continuous blood filtration, regulation of acid-base homeostasis and electrolyte balance, reabsorption of nutrients, and blood pressure control. As SIRT1 is an essential metabolic energy status sensor, its activity is regulated dynamically to respond to any alteration in the cellular metabolic state [14]. These adaptations allow cells to sense changes in energy levels in the nucleus, cytoplasm, and mitochondria. Therefore, the regulatory network of SIRT1 is composed of nutritional, hormonal, and environmental signals, SIRT1 target proteins, and NAD^+^ levels [13].

In this article, we review the protective effects of SIRT1 and NAD^+^ metabolism on renal function because these effects have been recently described based on several models of renal disorders with metabolic impairment. In addition, we provide an overview of the polyphenols that have the potential to restore SIRT1 and NAD^+^ metabolism in renal diseases. The protective effects of SIRT1 on the kidney include the maintenance of the glomerular barrier function, the antifibrotic effects, the antioxidant stress effects, and the regulation of mitochondrial function and energy metabolism [11,12,13,14], which are described in detail below.

## 2. SIRT1 Expression, Activity, and NAD^+^ Metabolism in the Kidney

SIRT1 is abundantly expressed in the kidney, and its expression decreases upon intake of a high-fat diet (HFD) or a high-glucose diet (HGD), while its expression increases during starvation and nutrient deprivation [15]. The importance of SIRT1 in renal tissue was emphasized in SIRT1-null mice that developed nephrogenic diabetes insipidus, characterized by the production of large amounts of dilute urine due to the inability of the kidney to concentrate it [16]. In contrast, the transgenic overexpression of SIRT1 in mice or cultured cells provides direct evidence of the protective potential of this sirtuin. Recently, it was demonstrated that upregulation of SIRT1 signaling in the early postnatal ages can attenuate the expression of lipogenesis, oxidative stress, and inflammatory markers, reversing some of the pathological mechanisms of kidney programming specifically affected by maternal obesity [17]. Similarly, SIRT1 transgenic mice demonstrate ameliorated diabetic kidney disease (DKD) because of reduced levels of oxidative stress markers [18], while induced SIRT1-deficient mice show elevated kidney inflammation [19] and urinary albumin excretion [18].

In contrast to other metabolically active organs, the kidney is composed of diverse cell types with very precise functions, in which SIRT1 plays specific roles. Some early studies showed that SIRT1 is mainly expressed in proximal tubule cells (PTCs) [6,20]. In recent years, single-cell transcriptome analysis showed that SIRT1 is expressed in the cell subpopulations of Bowman’s capsule, particularly in podocytes and proximal and distal convoluted tubules, as well as in principal and type B intercalated cells of the collecting duct and, to a lesser extent, in type A intercalated cells [21,22,23]. Furthermore, SIRT1 activity and expression gradually decrease in the kidneys of old mice compared to those of younger animals and are induced in metabolically stressful environments such as those generated by nutrient or oxygen deficiency [24,25,26]. In general, SIRT1 exerts a protective effect in the kidney by reducing inflammation, oxidative stress, blood pressure, and diabetic albuminuria and by delaying fibrogenesis and renal aging. These beneficial effects of SIRT1 in the kidney have been described by several authors [8,27,28] and are due to the deacetylation of specific lysine residues in several signaling network agents and transcription factors, including, among others, nuclear factor kappa-light-chain-enhancer of activated B cells (NF-κB); signal transducer and activator of transcription protein (STAT); forkhead box transcription factors (FOXOs); p38 mitogen-activated protein kinases (p38 MAPK); tumor protein p53 (p53); 5’-adenosine monophosphate-activated protein kinase (AMPK); mechanistic target of rapamycin (mTOR); and peroxisome proliferator-activated receptor-gamma coactivator 1 alpha (PGC1α). Recently, hypoxia-inducible factor (HIF)-2α and Beclin1 were added to the list of SIRT1-target proteins with implications in acute kidney injury (AKI); these are described in later sections of this review [29,30]. (See Table 1 and Figure 1).

Furthermore, SIRT1 activity is dependent on NAD^+^ levels. NAD^+^ is a coenzyme required for DNA stability and repair and for several oxidation–reduction reactions in the cell and is essential for energy generation through carbohydrate and lipid metabolism. For a long time, it was believed that the NAD^+^ concentration was stable in cells, changing only the cellular oxidation status. However, it is currently known that NAD^+^ levels decrease dramatically because of the activity of NAD^+^-consuming enzymes such as sirtuins, poly(ADP-ribose) polymerase (PARP), and ADP ribosyl synthetases, as well as chronically because of aging and altered nutrition-related conditions such as obesity and diabetes, contributing to the metabolic changes in these conditions [70]. The decrease in NAD^+^ levels is attributed to a decrease in the metabolic pathways involved in NAD^+^ synthesis. NAD^+^ can be synthesized from nicotinic acid (NA) through the Preiss–Handler Pathway, from tryptophan through de novo biosynthesis, and from nicotinamide or nicotinamide riboside (NR) through the NAD^+^ salvage pathway. Descriptions of these pathways can be found in a detailed review by Houtkooper et al. [71]. The de novo NAD^+^ biosynthetic pathway is particularly relevant in the kidney as this organ expresses all the NAD^+^ biosynthetic enzymes, such as indoleamine 2,3-dioxygenase (IDO) and quinolinate phosphoribosyltransferase (QPRT) [72]. Notably, the kidney also expresses α-amino-β-carboxymuconate-ε-semialdehyde decarboxylase (ACMSD), which forms α-aminomuconate-ε-semialdehyde (AMS) that can be oxidized to CO_2_ and H_2_O, decreasing the NAD^+^ synthesis and NAD^+^ levels. Therefore, the genetic and pharmacological inhibition of ACMSD increases NAD^+^ levels, improving mitochondrial and kidney function (Figure 2) [73].

ACMSD expression is restricted to the kidney, liver, and brain [74]. ACMSD inhibitors have been developed to increase de novo NAD^+^ biosynthesis and thus delay alterations in the mitochondrial function associated with the kidney’s physiological alterations. Notably, 3-[[[5-cyano-1,6-dihydro-6-oxo-4-(2-thienyl)-2-pyrimidinyl]thio]methyl]phenylacetic acid (TES-1025) was the first potent and selective inhibitor of human ACMSD enzymatic activity [73], and both in vitro and in vivo TES-1025 treatment in acute kidney disease leads to a concomitant increase in NAD^+^ levels and improved kidney function, as we describe below (Figure 2). In the following sections, we describe the mechanisms by which SIRT1 and the modulation of NAD^+^ metabolism are involved in the pathophysiology of kidney diseases [12,74,75], with special emphasis on SIRT1 alterations in AKI, chronic renal failure, obesity, and DKD.

### 2.1. The Role of SIRT1 in Acute Kidney Injury

Several clinical studies have suggested that AKI is an important risk factor for progressive chronic kidney disease (CKD), in which SIRT1 plays a pivotal role [75]. AKI includes a heterogeneous group of conditions in which renal proximal tubules are severely affected and unable to maintain electrolyte, acid-base, and fluid homeostasis due to an elevated metabolic rate caused by toxins or hemodynamic variations, leading to a rapid decline in renal excretory function. AKI manifests by increased serum creatinine levels and/or decreased diuresis and electrolyte abnormalities and/or coagulopathy, which may contribute to multiorgan failure [76,77,78,79]. AKI affects 3–7% of hospitalized patients, and patients with AKI have a high mortality rate [73]. AKI is associated with abnormal renal oxygen supply; therefore, some nephron segments turn to anaerobic metabolism. Alterations in ATP synthesis due to deficient metabolic pathways or lack of fuel substrates may increase cellular susceptibility to damage and dysfunction in the kidney [80]. Mechanisms involved in AKI in addition to alterations in energy metabolism are apoptosis, inflammation, autophagy, and cell cycle dysregulation [81] (Figure 1).

Cisplatin is widely used in the treatment of cancerous tumors. However, cisplatin induces cytotoxicity in renal tubules by inducing apoptotic cell death following DNA damage [82] and is frequently used to induce AKI in murine models. Furthermore, cisplatin induces peroxisome damage, which increases oxidative stress by decreasing catalase expression and increasing reactive oxygen species (ROS) production [26,83]. Notably, increasing SIRT1 expression specifically in the kidney proximal tubules attenuated cisplatin-induced AKI by preserving the peroxisome number and function, maintaining catalase activity, and preventing ROS production [84]. Additionally, aged animals subjected to short-term calorie restriction and cisplatin treatment presented with attenuated kidney injury associated with increased SIRT1 expression [85]. Under these conditions, SIRT1 deacetylates the tumor suppressor gene p53 and the transcription factor NF-κB p65 to maintain peroxisome integrity, induce catalase activity, and decrease the apoptosis rate and oxidant effect level, improving the glomerular filtration rate and protecting kidney function [82,84,86,87]. Recently, it was shown that fibroblast growth factor 21 (FGF21) exerts a renal-protective effect that is potentially mediated through SIRT1 activity (Figure 2). FGF21 increased the SIRT1 expression in cisplatin-treated mice and decreased blood urea nitrogen (BUN). The tissue structure presented fewer necrotic sites and the downregulation of apoptotic genes, effects that were not observed in cells when SIRT1 was genetically downregulated [88]. Melatonin is another hormone that exerts renoprotective effects through SIRT1. Melatonin activated the nuclear factor E2-related factor 2/heme oxygenase-1 (NRF2/HO-1) signaling pathway [89], decreased the HIF-1α content in human proximal tubular cells [90], and ameliorated AKI in severely burned rats [91]. However, research is still needed to evaluate the mechanisms by which SIRT1 mediates renoprotective effects.

It has been demonstrated that an integrative network of microRNAs and the transcriptome changes during AKI [92]. In fact, the decreased expression of SIRT1 has been associated with an increase in miR-183–3p. Remarkably, the downregulation of miR-183 during AKI in a rat model reestablished the SIRT1 expression and attenuated renal tissue damage [67]. Similarly, miR-34a and miR-155–5p reduced the expression of SIRT1 [93,94,95]. Taken together, these results suggested that the downregulation of SIRT1 is mediated through a posttranscriptional mechanism mediated by miRNAs (Figure 2). However, further research is needed to clarify whether these miRNAs bind directly to SIRT1 mRNA. Additionally, other miRNAs have been indirectly associated with the modulation of SIRT1 expression. For example, during lipopolysaccharide (LPS)-induced kidney injury, the miR-23 was reduced and associated with an increase in Rho-associated kinase 1 (ROCK1). ROCK1 suppressed SIRT1, leading to the activation of NFκB [96].

Among the mechanisms by which SIRT1 mediates renoprotective effects, a recent study showed that SIRT1 conditional knockout (KO) in interstitial cells led to severe renal damage and fibrosis, probably due to an increase in HIF-2α, which SIRT1 deacetylates in vitro. HIF-2α deacetylation is associated with a decrease in its own expression [30]. In addition, SIRT1 attenuated sepsis-induced AKI by activating Beclin1 deacetylation-mediated autophagy [29]. The modulation of kidney SIRT1 activity may also be a result of crosstalk communication with other types of tissues. Treatment with exosomes from mesenchymal stem cells in adipose tissue activated SIRT1 and decreased the state of inflammation, apoptosis rate, and microcirculation disorders in mice with sepsis-induced AKI [97].

On the other hand, the normal and proper function of the kidney depends greatly on nephron integrity, which requires high levels of energy and oxygen consumption to facilitate the active transport that leads to ion reabsorption and to produce glomerular filtrate. After the heart cells, renal cells have the highest density of mitochondria [98,99,100]. Recent evidence suggested that mitochondria are important in the pathogenesis of AKI, where several cellular pathways could be affected, increasing oxidative stress and leading to cytoskeletal changes, as well as the necrosis and apoptosis of renal tubular cells [101,102,103]. The SIRT1-activating compounds upregulate SIRT1 expression to reduce cisplatin- or hemorrhagic-shock-induced AKI, directly impacting catalase expression, eliminating free radicals, and promoting ATP generation [82,84,104]. SIRT1 also stimulates the activation of PGC1α, which is a transcriptional coactivator involved in energy metabolism and is the master regulator of mitochondrial biogenesis. Thus, stimulation of PGC1α restores the proximal tubule function by increasing mitochondrial content, which directly affects ATP synthesis [105,106,107,108]. SIRT1-activating compounds have also been proven effective during aging. Pretreatment with SIRT1-activating compounds decreased the vulnerability of aged kidneys to acute injury and revealed the clear progression of renal tubular injury after ischemia/reperfusion (I/R) in adult mice [6,109]. Altogether, these studies highlight the protective effect of SIRT1 activation mediated by bioactive compounds to suppress cell apoptosis and fibrosis [84,109], and we address this topic in detail.

Another condition that triggers AKI is metabolic endotoxemia, in which a high concentration of lipopolysaccharides (LPSs) accumulate in the intestinal lumen, through which they reach the circulatory system. Although the kidney metabolizes and excretes toxins, renal tubules can reabsorb a high concentration of LPSs, affecting the structure and function of the renal tubular epithelial cells and contributing to the development of AKI [110,111,112]. LPSs are recognized by Toll-like receptor 4 (TRL4), which undergoes conformational changes upon its activation, which in turn leads to the NF-κB signaling pathway activation and inflammatory cytokine production critical for pathogen detection and stimulation of innate immunity [113,114]. A study with TLR4-KO mice showed that the S1 segment of the proximal tubule exhibits the highest uptake of LPS due to a higher expression of TRL4. Notably, this segment is protected from oxidative stress because of an increase in SIRT1. In contrast, the S2 segment of the proximal tubule does not show increased SIRT1 expression and presents severe structural and functional damage caused by the deleterious effects of free radicals. This study suggests that, in the S1 segment, SIRT1 acts as a protective mechanism against sepsis-induced AKI [110]. On the other hand, SIRT1-activating compounds attenuated AKI by restoring SIRT1 expression in different animal models [19,112,115]. In summary, SIRT1 downregulates the expression and/or release of inflammatory mediators to protect against endotoxemia-induced AKI [19,112,116].

Regarding hemodynamic variations and AKI, SIRT1 has been implicated in cellular protection against renal I/R injury. In a recent study with an AKI rat model, the SIRT1 was upregulated after I/R treatment. The expression of SIRT1 in vitro was also elevated in human renal proximal tubular epithelial cells (HK2 cells) subjected to hypoxia and reoxygenation. In both the rat and the cell models, the pharmacological activation of SIRT1 significantly restored renal function and decreased the apoptosis rate, and it decreased the production of ROS in the HK2 cells. This study found that autophagy was upregulated in I/R injury and increased by pharmacological intervention. Thus, SIRT1 is crucial to renoprotection during AKI [117].

### 2.2. The Role of SIRT1 in Chronic Kidney Disease

Chronic kidney disease (CKD) is characterized by persistent urine anomalies and a gradual loss in kidney structure and function (e.g., a glomerular filtration rate <60 mL/min/1.73 m^2^ and/or albumin levels ≥30 mg per 24 h) for more than 3 months, and tubulointerstitial fibrosis is the hallmark of CKD regardless of its etiology [118,119,120]. The incidence of CKD and renal fibrosis affects up to 15% of the worldwide population and one-half of senior adults [121,122]. Therefore, proper diagnosis and patient surveillance are required to avoid adverse CKD-associated consequences, such as cardiovascular diseases and death. Among the risk factors for the development of CKD are low nephron count at birth, loss of functional nephrons due to aging, and acute or chronic kidney injuries caused by toxins or diseases [123], which frequently increase oxidative stress and decrease antioxidant activities.

Kidney fibrosis is the formation of excessive matrix connective tissue, whose components can cause kidney function deterioration [124]. Fibrosis is triggered as a beneficial reparative mechanism in response to initial damage. However, when one or more of the initial stages of fibrosis are incorrectly regulated pathological fibrosis develops and can progress to CKD. CKD fibrosis is associated with inflammation and tissue regeneration because of an inflammatory response involving tumor necrosis factor-alpha (TNF-α), inflammatory cytokines, the NLRP3 inflammasome, and transforming growth factor-beta (TGF-β), causing a gradual reduction in renal function [124,125]. Similarly, oxidative stress disables the kidney’s ability to enable damage repair, which leads to necrosis and apoptosis of the renal tissue cells [103,126,127]. SIRT1 safeguards cells from oxidative stress and inhibits the expression of profibrotic markers. For instance, this deacetylase can stimulate the expression of antioxidants via the FOXO pathways and inhibit NF-κB signaling, which is a major inducer of inflammatory responses [128].

After exposure to high-permeability-inducing and low-oxygen environments, SIRT1 protects the renal medullary interstitial cells from oxidative stress-induced apoptosis. Knocking down SIRT1 in models of unilateral ureteral obstruction (UUO), a model of CKD, significantly reduced oxidative stress resistance due to decreased expression of antioxidant enzymes, specifically cyclooxygenase-2 (COX-2), which led to increased fibrosis and apoptosis rates in the renal medulla, suggesting that SIRT1 plays a relevant role in the protection of kidney cells against oxidative stress [6,129]. Furthermore, pharmacological activation of SIRT1 has been shown to prevent the characteristic alterations in CKD. For instance, treatment with the SIRT1 activator SRT1720 in UUO model mice increased COX-2 expression and attenuated renal fibrosis and inflammation [6,130]. Moreover, activation of SIRT1 attenuated renal inflammation and tubular fibrosis in 5/6 nephrectomized rats and cultured mesangial cells (MMCs) through inhibition of TGF-β/SMAD signaling [25,69,131,132]. In contrast, the sirtuin inhibitor sirtinol eliminated the renoprotective effect of losartan, which suppresses endoplasmic reticulum (ER) stress by upregulating SIRT1 via HO-1 and thioredoxin, in this UUO model. In general, SIRT1 is a key mediator of the antifibrotic effect of angiotensin II blockers [133] and hydrogen-rich water [134].

In particular, the relevance of SIRT1 in maintaining endothelial function was demonstrated with a SIRT1 endothelial-cell-deficient mouse model (SIRT1^endo-/-^) [135]. The kidneys of SIRT1^endo-/-^ mice showed a significant increase in tubulointerstitial fibrosis after folic acid treatment. Additionally, enhanced peritubular capillary rarefaction manifested as impaired angiogenesis and increased activity of the developmental Notch1 signaling pathway [136]. The increased activity is attributable to the suppressed expression of matrix metalloproteinase-14 (MMP-14) [135]. The restoration of MMP-14 improved endothelial proliferation and function, inhibited fibrosis, and ameliorated nephrosclerosis [135]. In the UUO and I/R models, SIRT1 activation improved renal fibrosis in proximal tubular cells by deacetylating SMAD3 and SMAD4 and blocking the effect of TGF-β signaling on matrix metalloproteinase-7 (MMP-7), which normally cleaves E-cadherin [25,137,138]. As a result of SIRT1 deacetylation of SMAD4, β-catenin remains attached to E-cadherin at the cell-cell junctions, thus reducing fibrosis [137]. In addition, SIRT1 exerts its antifibrotic role through modulation of PGC1α activity [108]. Taken together, these studies showed that SIRT1 is a therapeutic target whose activation attenuates the progression of renal fibrotic diseases (Figure 1).

SIRT1 is involved in the activation of FOXO1, FOXO3, and FOXO4 to regulate the cell response to inflammation, autophagy, apoptosis, and oxidative stress [139,140,141]. Autophagy, a lysosomal degradation process that is evolutionarily conserved, provides nutrients and energy to the cell by eliminating unessential material during starvation, leading to cellular renovation and homeostasis [142]. In the kidney, autophagy maintains cellular homeostasis and the viability of renal cells such as podocytes and tubular epithelial cells. Therefore, the impairment of autophagy is specifically linked to the pathogenesis of CDK. For instance, the activation of SIRT1-induced autophagy in the kidney exerts a protective effect, which is frequently diminished in the kidneys of aging animals subjected to systemic hypoxia and ischemic renal injury [140,143,144]. In contrast, autophagy deficiency in renal proximal epithelial tubular cells displays increased sensitivity against renal injury even under normal feeding conditions [145]. In fact, the activation of FOXO3 by SIRT1 is essential for the control of hypoxia-induced autophagy in proximal tubule cells [140]. Increased susceptibility to oxidative stress has been attributed to phosphoinositol-3 (PI3)-kinase activity and AKT phosphorylation, in addition to SIRT1, as they were significantly reduced in the kidneys of aging animals upon gradual exposure to hypoxia [146]. This mechanism is particularly relevant during aging. The kidneys of aged SIRT1-deficient mice under calorie restriction and hypoxia showed the accumulation of damaged mitochondria in addition to increased FOXO3 acetylation [140]. Therefore, as an adaptation to hypoxia, SIRT1 may mediate mitochondrial autophagy in the proximal tubule cells of senile kidneys subjected to energy restriction. Additionally, transgenic mice with the kidney proximal-tubule-specific overexpression of SIRT1 showed attenuated development of AKI because of decreased local oxidative stress and attenuated tubular cell apoptosis that had been induced by cisplatin treatment [84].

Inflammation also determines the initiation and progression of CKD, and, along with ischemic changes, it affects tissue oxygenation, the recruitment and activation of profibrotic and proinflammatory cytokines, and the growth factor activity and the subsequent production of harmful ROS [147,148,149]. Among the key signaling molecules involved in the control of inflammation, ubiquitously distributed NF-κB is rendered inactive in the cytoplasm under basal conditions and in response to various stimuli, such as high glucose, proteinuria, cytokines, and growth factors. NF-κB is translocated into the nucleus to turn on the transcription of several inflammatory, cell cycle, and apoptotic genes [150,151,152]. NF-κB is a heterodimeric complex consisting of a p50 and p65 subunit. SIRT1 interacts and deacetylates p65, inhibiting the transcriptional activity of NF-κB, leading to attenuate inflammation [31,153]. Additionally, experimental results demonstrated that, after LPS challenge, knocking down SIRT1 led to hyperacetylation and activation of NF-κB p65 [19]. In contrast, overexpression or pharmacological stimulation of SIRT1 prevented acetylation of NF-κB p65 and cytotoxicity in proximal tubule cells during cisplatin treatment [82,128,154]. Together, these findings demonstrated the key role of SIRT1 as a negative regulator of the NF-κB p65 signaling pathway and revealed that SIRT1 exerts a renoprotective effect by attenuating cell apoptosis and preventing inflammation (Figure 1).

Extracellular high-mobility group box 1 (HMGB1), or amphoterin, is another target of SIRT1 [155]. Under physiological conditions, hyperacetylated HMGB1 is located primarily in the nucleus, from where it is released into the cytoplasm upon the application of stress to nonlethally injured and necrotic cells, and then, it acts as a potent proinflammatory cytokine, inducing the systemic inflammation that contributes to the pathogenesis of various diseases [156,157,158]. SIRT1 participates in regulating the nuclear retention of HMGB1 by deacetylating it [155]. The elimination of SIRT1 increases HMGB1 acetylation, facilitating its nuclear-to-cytoplasmic translocation and systemic release, which increases cellular damage. In contrast, the pharmacological activation of SIRT1 promotes the deacetylation of HMGB1 and its nuclear retention, thus preventing renal damage [155]. Therefore, under stress conditions, SIRT1 modulates the damage signaling initiated by HMGB1 secretion by forming an anti-inflammatory complex with HMGB1, reducing renal cell damage.

### 2.3. Modulation of NAD^+^ Metabolism Improves Renal Function in Acute Kidney Injury and Chronic Kidney Disease

The use of the ACMSD inhibitor has proven to be efficient in protecting renal function in two different models of AKI: one induced by a single intraperitoneal (i.p.) application of cisplatin and the other induced by ischemia/reperfusion (I/R) induction achieved by clamping renal pedicles for 25 min. Dietary treatment with the ACMSD inhibitor TES-1025 protects against AKI by normalizing the blood creatinine, urea nitrogen, and kidney injury molecule (KIM)-1 levels and the glomerular filtration rate (Figure 2). The cumulative histopathological score used to evaluate tubular necrosis, tubular dilation, inflammation, edema, and cast formation was unchanged by the ACMSD inhibitor treatment. Moreover, alterations in NAD^+^ levels have been observed in different chronic diseases, and they were accompanied by alterations in the ACMSD expression. For example, the ACMSD expression increases in streptozotocin (STZ)-induced diabetic rats [159]. In kidney biopsy samples obtained from patients with CKD, the expression of enzymes involved in NAD^+^ de novo synthesis decreased in the immediate post-transplantation period, while the salvage pathway was stimulated. Interestingly, nicotinamide riboside (NR) supplementation in mouse models of AKI prevented tubular injury, but not in CKD [159]. A second study using high doses of i.p. nicotinamide showed the dose-dependent acquisition of the protective phenotype against tubulointerstitial fibrosis [160]. However, further research is needed to establish the reason why the compelling effects observed in AKI are not observed in CKD. It is possible that a longer and/or earlier treatment may be required or that the administration route and dose may be determinants of the effects observed. Additionally, translational studies are required to evaluate whether the effects observed in mouse models are replicated in humans. The first evidence obtained through a randomized, double-blind, placebo-controlled study in hospitalized patients with AKI showed that AKI resulted in a 50% reduction in whole blood NAD^+^, and strikingly, the use of increasing doses of NR and pterostilbene augmented these NAD^+^ levels by 37% within 48 h, although no difference was observed in the kidney function evaluation measures [161]. A previous study with 12 healthy volunteers showed that NR alone, at doses ranging from 100 to 1000 mg, increased peripheral blood mononuclear cell NAD^+^ levels after 24 h [162]. The compounds were well tolerated, and no safety concerns were raised. Together, these results revealed the therapeutic potential of NR supplementation to modulate NAD^+^ levels in patients with kidney disease. Nevertheless, the effect of NR supplementation on kidney NAD^+^ levels and renal physiology needs more investigation.

From a mechanistic point of view, extracellular signal-regulated kinase 1/2 (ERK1/2) was activated during AKI, inducing renal dysfunction after I/R, and this change was associated with a decrease in mitochondrial biogenesis. The inhibition of mouse ERK1/2 activation in AKI using the mitogen-activated protein kinase 1/2 (MEK1/2) inhibitor trametinib attenuated renal cortical NAD^+^ depletion. The expression of the rate-limiting NAD^+^ biosynthesis salvage enzyme nicotinamide phosphoribosyl transferase (NAMPT) was decreased following AKI induction, and this decrease was prevented by ERK1/2 inhibition. The microRNA miR-34a level decreased with the inhibition of ERK1/2, leading to an increase in NAMPT protein. The mice treated with an miR-34a mimic showed no increase in NAMPT protein in the renal cortex after ERK1/2 inhibition. In addition, ERK1/2 activation increased the level of acetylated PGC1α, the less active form, whereas inhibition of ERK1/2 activation prevented an increase in acetylated PGC1α after AKI induction because of the SIRT1 and NAD^+^ attenuation [163].

### 2.4. Alterations in SIRT1 Activity in Obesity and Diabetes

SIRT1 deacetylase activation delays and/or attenuates many obesity-related diseases, including nonalcoholic fatty liver disease (NAFLD) and diabetes, and has received great attention as a drug target. SIRT1 function is aberrantly low in obesity; therefore, understanding the underlying mechanisms is important for drug development. Additionally, the dysregulation of metabolic pathways involving SIRT1 may lead to type 2 diabetes (T2D), which is characterized by high glucose concentrations caused by insulin resistance. The development of T2D and its complications is associated with lipid metabolism disorders. Inadequate fatty acid oxidation gives rise to the initiation of insulin resistance and lipid accumulation [164,165]. In the following sections, we describe the role of SIRT1 in delaying the alterations observed in the kidney during obesity and diabetes.

#### 2.4.1. The Role of SIRT1 during Obesity-Induced Renal Lipid Accumulation

Obesity is a global public health threat mainly characterized by hyperinsulinemia, altered glucose and lipid metabolism, chronic inflammation, and oxidative stress and is a risk factor for the development of cardiac disorders, metabolic syndrome, and T2D [166]. Obesity-induced metabolic alterations exert profound effects on kidney physiology, including tubular and glomerular hypertrophy, thickening of the glomerular basement membrane, and mesangial matrix expansion [167], which are mainly caused by lipid deposition, chronic inflammation, an increased glomerular filtration rate, and other alterations [168]. This kidney pathological damage contributes to albuminuria and eventually to tubulointerstitial fibrosis and glomerulosclerosis [169,170]. Notably, direct evidence of the potential of SIRT1 modulation in improving kidney injury upon HFD intake was obtained from transgenic mice overexpressing SIRT1 and mice treated pharmacologically with SRT1720 [17,171,172]. SIRT1 overexpression and SRT1720 treatment attenuated the renal lipid content and the expression of lipogenesis, oxidative stress, and inflammatory markers [17]. Furthermore, the effects of an HFD on renal physiology were transgenerational, as the male offspring of high-fat-fed mouse dams exhibited perirenal fat deposition with lower SIRT1 expression and reduced AMPK phosphorylation [173,174].

As the body ages and body weight increases, vascularization decreases, leading to a lower tissue oxygen supply and, ultimately, to chronic hypoxia [174,175,176]. Persistent hypoxia causes changes in the mitochondrial functions, which shift from oxidative stress to anaerobic glycolysis [177]. Proximal tubule cells are easily damaged by hypoxia because their volume accounts for more than 80% of the kidney cortex, to which oxygen is supplied only by peritubular capillaries [178]. The most characterized effectors of low-oxygen responses are the hypoxia-inducible factors HIF-1 and HIF-2, which facilitate oxygen delivery and cellular adaptation to hypoxia [179]. In the kidney, HIF-1α is located in the fully developed organ, and HIF-2α is expressed in the renal endothelial cells as well as in the interstitial fibroblast-like cells [180]. Under normoxia, SIRT1 binds and deacetylates HIF-1α. In response to acute hypoxia, SIRT1 activity decreases, allowing the acetylation and activation of HIF-1α, which becomes stable [62,181]. HIF-2α is stimulated by chronic hypoxia [182,183] and plays a regulatory role that opposes the detrimental proinflammatory effects of HIF-1α [182].

During aging, SIRT1 stimulates mitochondrial biogenesis and regulates nuclear-mitochondrial communication by deacetylation/inactivation of HIF-1α, which is disrupted under pseudohypoxia and low levels of NAD^+^ [183]. A recent study of kidneys from aged mice showed a decrease in SIRT1 expression and an increase in the acetylated/active form of HIF-1α, along with a greater accumulation of the extracellular matrix and a higher apoptosis rate compared to kidneys from young mice. Additionally, in HK2 cells exposed to hypoxic conditions, SIRT1 was downregulated, allowing the acetylation/activation of HIF-1α. SIRT1 overexpression in HIF-1α-transfected HK2 cells under hypoxia confirmed the inhibition of HIF-1α activity by the SIRT1-induced deacetylation of HIF-1α. The chronic activation of HIF-1α increased the apoptosis and fibrosis rates, suggesting that SIRT1-induced deacetylation of HIF-1α might confer protection against tubulointerstitial damage in aged kidneys [65].

The role of SIRT1 in obesity-induced renal damage has begun to be recognized through studies on the molecular mechanisms produced by the currently used pharmacological treatments. For example, liraglutide is a pharmacological treatment for patients with T2D that reduces the events associated with persistent microalbuminuria, serum creatinine, and end-stage kidney disease [184]. In this regard, liraglutide has been shown to increase SIRT1 expression in the kidneys of HFD-fed mice and prevent apoptosis in a SIRT1-dependent fashion in vitro [185]. Moreover, liraglutide reduced renal lipids increased the number of mitochondrial tricarboxylic acid (TCA) metabolites and the NAD^+^ level and increased the expression of SIRT1, AMPK, and PGC1α in the kidneys of HFD-fed rats [186]. Similarly, elafibranor, a dual peroxisome proliferator-activated receptor alpha (PPARα) and PPAR-delta (PPARδ) agonist, increased the renal expression of PPARα, PPARδ, SIRT1, autophagy genes, and glomerular/renal tubular barrier markers and was associated with low renal tubular injury and tubulointerstitial fibrosis scores in HFD-fed mice. Moreover, acute treatment of podocytes and HK2 cells with elafibranor or recombinant SIRT1 reversed the oxidative stress, autophagy dysfunction, cell apoptosis, barrier marker loss, and albumin endocytosis induced in vitro by serum obtained from HFD-fed models [187].

SIRT1 activity may be restored by decreasing renal lipid accumulation. In fact, the treatment of diabetic mice with CL316,243, a β3-adrenergic receptor agonist, activated brown adipose tissue, which decreased the blood lipid profile and improved renal fibrosis, inflammation, and oxidative stress. Additionally, CL316,243 increased the renal expression of AMPK, SIRT1, and PGC1α [188]. Furthermore, exercise training of diabetic mice decreased triglyceride levels, reduced kidney morphological alterations, prevented the loss of SIRT1, improved mitochondrial function, and reduced the acetylation of NF-κB in the kidney compared to the diabetic mice without exercise training [189]. However, more research is needed to understand the hormonal and metabolic pathways involved in this response.

#### 2.4.2. The Role of SIRT1 in Diabetic Kidney Disease

Diabetes consists of metabolic disorders with multiple etiologies characterized by chronic hyperglycemia. Globally, diabetes prevalence is increasing significantly and is considered a major health issue of alarming proportions. As of 2019, nearly half a billion people were living with diabetes worldwide, and this number is expected to increase in the future [190]. Untreated diabetes disrupts carbohydrate, fat, and protein metabolism as a result of defects in insulin management, increasing the production of ROS and leading to a state of chronic oxidative stress and the generation of inflammatory mediators, as well as vascular changes and dysfunction that lead to extensive morbidity and ultimately death [5]. The long-term vascular complications caused by diabetes are cardiovascular diseases, such as retinopathy, neuropathy, and nephropathy [5,191,192].

Approximately 30% of diabetic patients develop diabetic nephropathy, also known as diabetic kidney disease (DKD), which is a leading cause of the progression to end-stage renal disease (ESRD) [192,193]. Typically, DKD patients display albuminuria and a low estimated glomerular filtration rate (eGFR) due to glomerular and tubular damage [23,170,192,194,195]. Oxidative stress, mitochondrial dysfunction, apoptosis, autophagy, and inflammation are some of the mechanisms involved in the kidney damage sustained in DKD [195,196]. Remarkably, in DKD, SIRT1 activation exerted a protective effect on the kidney in several models of renal injury [18,197,198,199,200], whereas the elimination of SIRT1 exacerbated renal alterations in diabetes [18,19].

SIRT1 deacetylates FOXO4, decreasing the expression of the proapoptotic gene Bcl2l11, preventing podocyte loss in diabetes [201]. SIRT1 also inactivates NF-κB and STAT3 to attenuate proteinuria and podocyte injury [202]. Interestingly, the podocyte-specific loss of SIRT1 (SIRT1^pod−/−^) decreased the podocyte numbers and worsened albuminuria, which in turn accelerated renal disease progression in diabetic mice [201,202,203]. Podocytes obtained from SIRT1^pod-/-^ mice exhibited severe morphological changes, such as enhanced podocyte foot process effacement and reduced the expression of nephrin and nestin. SIRT1 is required for the preservation of cytoskeletal integrity and podocyte survival through deacetylation and the concomitant activation of the actin-binding and actin-polymerizing protein cortactin [204,205]. Moreover, the reduced expression of SIRT1 in podocytes plays an important role in age-induced kidney injury, aggravating glomerulosclerosis and albuminuria, as indicated by the increased expression of cellular senescence markers [203]. In contrast, inducible podocyte-specific SIRT1 overexpression in type 1 diabetic OVE26 mice attenuated the progression of diabetic glomerulopathy by reducing albuminuria and attenuating diabetes-induced podocyte loss and oxidative stress. Indeed, the podocytes treated with a pharmacological SIRT1 agonist, both in vivo and in vitro, showed increased deacetylation and activation of PGC1α, which protected these cells against high-glucose-mediated mitochondrial injury and effectively mitigated the progression to diabetic kidney disease induced by oxidative stress [198]. Furthermore, SIRT1 coordinates functional interplay between the podocytes and tubular cells in DKD [18]. SIRT1^pod-/-^ mice presented severe mesangial expansion and cell loss [202], while SIRT1 overexpression reduced renal damage in diabetic animals [198]. In STZ-induced diabetic animals, the elimination of SIRT1 in proximal tubules upregulated the tight junction protein claudin-1, which led to exacerbated albuminuria and impaired renal function [18].

Recent studies have demonstrated that SIRT1 not only participates in the functional interplay between proximal tubule cells and podocytes but also associates with sodium-glucose cotransporter 2 (SGLT2). High basolateral glucose levels in the proximal tubules of diabetic kidneys might upregulate the apical SGLT2 to increase glucose uptake through the basolateral glucose transporter 2 (GLUT2), triggering a simultaneous reduction in SIRT1. Diabetic mice and hyperglycemia-exposed proximal tubule cells treated with the SGLT2 inhibitor canagliflozin exhibited recovered SIRT1 expression both in vivo and in vitro, respectively [206]. SGLT2 inhibitors also activate AMPK, leading to an increase in cellular AMP or ADP levels [207]. As the SGLT2 inhibitors induce both AMPK and SIRT1, which are known to promote autophagy, they dramatically attenuate glomerular hyperfiltration, cell stress, and kidney injury. Hence, the interaction between the AMPK/SIRT1 signaling and the sodium transport mechanisms in the kidney may explain the protective effects of SGLT2 inhibitors in the development of DKD [208].

The evidence indicated that SIRT1 may interact with the renin-angiotensin system (RAS). SIRT1 activates the angiotensin-converting enzyme 2 (ACE2) promoter, stimulating the production and positive effects of angiotensin 1–7 [Ang-(1–7)] [209,210]. Ang-(1–7) increases SIRT1 expression, whereas angiotensin II (Ang II) expression has the opposite effect. The podocytes exposed to Ang II showed a reduced expression of SIRT1 in association with acetylation of p53. Diabetic mice treated with angiotensin receptor blockers, which are known to attenuate albuminuria and prevent podocytes from undergoing apoptosis, were associated with increased SIRT1 activity and reduced p53 acetylation in the kidney [211].

In mesangial cells, SIRT1 deacetylates and inhibits the actions of NF-κB, preventing the phosphorylation-induced activation of cJUN, decreasing matrix expansion and attenuating proteinuria [212]. SIRT1 also inhibited renal inflammation and fibrosis through the deacetylation of HIF-1α under high glucose conditions [213]. SIRT1 inhibits the activation of AKT and increases the level of AMPK signaling, which prevents the induction of mTOR-mediated protein synthesis and the subsequent mesangial hypertrophy induced by high glucose [214]. Additionally, SIRT1 can protect against advanced glycation end product (AGE)-induced diabetic renal fibrosis through its antioxidative effect [197]. Treatment of the mesangial cells with AGEs reduced SIRT1 protein expression and activity but induced the overproduction of fibronectin and TGF-β1 levels in a dose- and time-dependent manner. SIRT1 upregulates the expression of NRF2 and activates the antioxidant response element (ARE) antioxidative pathway to prevent the progression of renal fibrosis [69,215]. The expression of proinflammatory and profibrotic cytokines in vitro and in vivo was reported in the HBZY-1 rat glomerular mesangial cell line treated continuously in a high-glucose milieu and in a diabetic mouse model. Similar results were observed in both models, which showed that SIRT1 expression was significantly reduced in a high-glucose environment, whereas the expression of NF-κB and FOXO1 was increased, and renal cell fibrotic factors, such as vimentin and fibronectin, were induced [216].

On the other hand, connexins (Cxs) are transmembrane proteins that exchange information between adjacent cells. The most abundantly expressed and widely distributed gap junction protein is connexin 43 (Cx43), whose expression is significantly decreased in DKD patients and diabetic animal models [217]. It has recently been shown that Cx43 is important in regulating the renal epithelial-to-mesenchymal transition (EMT) and diabetic renal tubulointerstitial fibrosis through the SIRT1/HIF-1α signaling pathway, as previously described [218]. Moreover, higher levels of glucose induce extracellular matrix accumulation; the upregulation of poly (ADP-ribose) polymerase-1 (PARP-1), which is essential for initiating various forms of DNA repair [219]; and a decrease in SIRT1 expression. The decreases in SIRT1 can be attributed to a decrease in the NAD^+^ levels caused by PARP activation, which inhibits the AMPK/PGC1α signaling pathway, inducing DKD pathogenesis [220].

## 3. Natural Modulators of SIRT1 Activity in the Kidney

The evidence obtained through studies with mouse models showed that the activation of SIRT1 alleviated renal pathological abnormalities [221]. Therefore, measures have been taken to identify natural compounds that can activate SIRT1. The identification of SIRT1 modulators of natural origin, such as polyphenolic products in fruits, vegetables, and plants, demonstrated potential therapeutic effects in the prevention and treatment of kidney diseases. Polyphenols are plant secondary metabolites and represent a large group of compounds of variable structural complexity with aromatic rings containing one or more hydroxyl groups. Numerous reports are suggesting that polyphenols in food promote the enzymatic activity of SIRT1, thereby controlling gene expression in processes such as DNA repair, metabolism, the oxidative stress response, mitochondrial function, and biogenesis [222]. In addition, SIRT1 epigenetic modifications are also associated with genome stability, gene transcription, and metabolic regulation. Thus, polyphenols, through SIRT1 stimulation, can modulate the activation or silencing of specific genes. There are several categories of polyphenols, including those based on their origin, biological function, and chemical structure [223,224]. Their classification by chemical structure is probably the most widely adopted method, and through this categorization, polyphenols are classified into two main groups: flavonoids and nonflavonoids.

### 3.1. Nonflavonoid Polyphenols

#### 3.1.1. Stilbenes

Stilbenes are phenolic compounds found in various families of plants [225]. Some of these secondary metabolites have been recognized as phytoalexins and associated with the defense mechanisms of plants as they are produced after infection by pathogens or exposure to UV radiation and exhibit antifungal activities. The stilbene structure is based on the C6–C2–C6 backbone and is defined by two aromatic rings linked by an ethylene bridge [226] (Figure 3).

##### Resveratrol

Resveratrol (3,5,40-trihydroxy-trans-stillbene) is a natural plant polyphenol belonging to the family of stilbenes because it contains the two phenyl moieties connected by a two-carbon methylene bridge that is common to all the family members [227]. Resveratrol is found in the skin of grapes, berries, and peanuts and is a known activator of SIRT1 [59], which leads to antiproliferative and anti-inflammatory activities [45,228,229]. Resveratrol increases SIRT1 activity approximately tenfold [230]. Hou et al., using molecular dynamics simulations, provided an in silico model supporting that resveratrol modulates SIRT1 activity by stabilizing SIRT1/peptide interactions in a substrate-specific manner [231]. All the available in vitro and in vivo animal and human studies examining the effects of resveratrol showed that resveratrol improves renal tissue function after kidney alterations [231]. Those studies have demonstrated that SIRT1 expression is increased in the kidney by resveratrol. However, whether resveratrol modulates directly SIRT1 activity has not been evaluated, and it is not clear which of the resveratrol positive effects on the kidney are directly mediated by SIRT1. There is evidence that shows resveratrol indirectly activates SIRT1 by activation of AMPK. AMPK acts as a primary sensor that increases the NAD+ levels, thus inducing stronger deacetylation of the SIRT1 targets, due to a concomitant increase in SIRT1 activity [232,233]. We address those studies in the following sections (Figure 4).

##### SIRT1-Mediated Effects of Resveratrol in the Kidney Inflammatory Response, Oxidative Stress, and Fibrosis

Resveratrol modulates SIRT1 activity with a concomitant reduction in fibrosis, mesangial expansion, oxidative stress, and inflammatory cytokine levels while improving kidney structure and function [234,235]. Moreover, Saldanha et al. [236] found that the expression of the key proinflammatory factors mediated by NF-κB and STAT3 was increased in the kidneys of Sirt1 KO db/db mice, further confirming a key role of SIRT1 in the regulation of inflammation in the kidney. On the other hand, SIRT1 modulates the expression and activity of PGC1α [59], regulates AMPK activity through deacetylation and activation of serine/threonine kinase 11 (STK11)/liver kinase B1 (LKB1) [237], and controls the expression of manganese superoxide dismutase (MnSOD) [238]. Resveratrol has been associated with an increase in all these key SIRT1 target proteins, which are essential in maintaining mitochondrial function and reducing oxidative stress [239] (Figure 4).

##### SIRT1 Activation by Resveratrol in Different Kidney Cells

The high glucose exposure of primary mesangial cells induces ROS production and causes a decrease in MnSOD activity; however, treatment of these cells with resveratrol reduces hyperglycemia-induced ROS alterations [240]. Additionally, resveratrol treatment prevents the reduction in the mitochondrial complex III activity induced by exposure to high glucose, which is an important source of mitochondrial oxidative stress. Moreover, treatment with resveratrol restores mitochondrial membrane potential hyperpolarization, which is harnessed for ATP production, and preserves mitochondrial DNA content [240]. Importantly, because of the resveratrol administration, the mesangial cell damage induced by hyperglycemia-induced oxidative stress was directly alleviated by stimulating the SIRT1 activity [240], indicating an important role for SIRT1 in mediating the effects of resveratrol.

In a study conducted by Hui et al., in which SV40 MES 13 mouse mesangial cells were treated with resveratrol, the TGF-β1-induced mitochondrial function was attenuated, the mitochondrial membrane potential and ATP production levels were increased, and the ROS production was inhibited [241]. In addition, resveratrol improved mitochondrial function by increasing complex I/III activities and fission/fusion protein expression [241], indicating a protective effect of resveratrol on the electron transport chain. Moreover, SIRT1 has been found to inhibit TGF-β signaling by individually deacetylating either SMAD3 [69] or SMAD4 [138] in renal cells. Interestingly, in an unexpected finding, resveratrol interacted with both SMAD3 and SMAD4 simultaneously to prevent the formation of the SMAD3/4 complex via SIRT1, inhibiting SMAD3 phosphorylation and renal fibrosis [242].

Resveratrol treatment also increased the SIRT1 protein levels, while the PGC1α and acetylated-PGC1α protein levels were reduced, suggesting that the protective effects of resveratrol on mitochondria may be linked with SIRT1 activation [241].

Interestingly, Uchida et al. demonstrated the potential proinflammatory effects of resveratrol treatment (50–75 µM) on glomerular mesangial cells [243]. Glomerular mesangial cells co-stimulated with the proinflammatory cytokine interleukin-1 (IL-1) or TNFα and subsequently treated with resveratrol showed increased NF-κB activity [243]. This increased proinflammatory effect was also shown in the LLCPK1 kidney proximal tubule cells. However, when the glomerular mesangial cells and LLCPK1 cells were treated with resveratrol alone, no significant changes in NF-κB occurred, suggesting that resveratrol exerts proinflammatory effects only in the presence of preexisting proinflammatory cytokines (Figure 4).

Yang et al. showed that the yin yang 1 (YY1) protein plays an important role in renal fibrosis during DKD [244]. Interestingly, Du et al. explored the underlying mechanisms that regulate the initiation of the EMT within the kidney tubules in diabetes and discovered that SIRT1 activation by resveratrol deacetylated YY1 and inhibited the EMT in DKD [245]. Another study with rat epithelial cells demonstrated that resveratrol stimulated SIRT1 to deacetylate SMAD3, which reversed the TGF-β1-induced upregulation of type IV collagen and fibronectin mRNA levels to attenuate renal fibrosis [25]. Kim et al. demonstrated that activation of SIRT1 by resveratrol reduced cisplatin-mediated p53 acetylation and ameliorated cisplatin-induced kidney injury by inhibiting the apoptotic pathway [82]. Furthermore, Xiao et al. found that SIRT1 upregulation by resveratrol was critical for the downstream blockade of the EMT process in HK2 cells because it increased the E-cadherin level. In addition, profibrotic MMP-7, α-smooth muscle actin (α-SMA), and type I alpha 1 collagen mRNA and protein levels were decreased in the resveratrol-treated group. The authors concluded that resveratrol attenuates renal injury and fibrosis by inhibiting the EMT process, and this inhibitory effect was attributed to the resveratrol-induced upregulation of SIRT1, which led to deacetylated SMAD4 and reduced MMP-7 expression [138]. Wang et al. showed that resveratrol modulated the SIRT1/FOXO3a pathway by increasing SIRT1 deacetylase activity, subsequently ameliorating hyperglycemia-induced renal tubular oxidative stress damage [246]. Liu et al. showed that low concentrations of resveratrol (5–20 μM) decreased the TGF-β1-induced EMT via the SIRT1-dependent deacetylation of the SMAD3/SMAD4 mechanism. In contrast, long-term (72 h) exposure to high concentrations of resveratrol (≥40 µM) promoted the EMT in HK2 cells via mitochondrial oxidative stress and ROCK1-mediated disordered cytoskeleton remodeling. In vivo, low-dose resveratrol treatment (≤25 mg/kg) partially improved renal function, whereas high-dose resveratrol treatment (≥50 mg/kg) led to loss of antifibrotic activity and even aggravated renal fibrosis [247]. Furthermore, Huang et al. examined the expression of SIRT1, which had been associated with renal protection in AKI [84]. The expression of SIRT1 and phosphorylated SIRT1 was not altered following treatment with ioxitalamate and/or resveratrol, indicating that SIRT1 did not play a role in ioxitalamate-induced cytotoxicity in HK2 cells [248].

Regarding podocytes, Zhang et al., conducted a study finding that resveratrol attenuated high-glucose-induced mitochondrial stress-triggered ROS production and increased the membrane potential [249]. In addition, resveratrol treatment increased the respiratory chain complex I and III activity levels, while the number of proapoptotic proteins released from the mitochondria were reduced, suggesting improved mitochondrial functioning and reduced podocyte damage. Additionally, SIRT1, PGC1α, nuclear respiratory factor-1 (NRF-1), and transcription factor A mitochondrial (TFAM) mRNA and protein levels were increased after resveratrol treatment [249] (Figure 4).

##### Resveratrol Protects against Diabetic Kidney Disease by Activating SIRT1

A study by Kim et al. demonstrated that DKD in *db/db* mice was ameliorated by resveratrol (20 mg/kg/day) treatment for 12 weeks, resulting in reduced lipotoxicity-related apoptosis of renal cells and decreased oxidative stress. Resveratrol treatment significantly reduced kidney nonesterified fatty acid (NEFA) and triacylglycerol contents and attenuated albuminuria [250]. In the mesangial area, the number of TGFβ1-, type IV collagen-, and F4/80-positive cells were significantly reduced after the resveratrol treatment, indicating attenuated modification of the renal diabetic phenotype. The resveratrol treatment increased phosphorylated AMPK, estrogen-related receptor-1α (ERR-1α), SIRT1, and PGC1α, while it decreased sterol regulatory element-binding protein 1 (SREBP1) levels. This study suggested that the activation of AMPK and SIRT1 by resveratrol may prevent the renal lipid accumulation and cell injury related to the activation of PGC1α [250]. A study conducted by Xu et al. showed that the pretreatment of rat mesangial cells with resveratrol 6 h before high-glucose treatment significantly reduced the hyperglycemia-induced increase in ROS production [240]. The beneficial effects of resveratrol were blocked by SIRT1-specific siRNA and the SIRT1 inhibitor EX-527. This study demonstrated that resveratrol protects mesangial cells against hyperglycemia-induced oxidative damage directly or indirectly through SIRT1 activity stimulation. It was further reported that resveratrol treatment protected against the development of DKD by preventing a decrease in the expression of SIRT1, and these results were consistent with the results of studies in which STZ was used to induce diabetes in rats [251]. Interestingly, a recent study reported that resveratrol protected against DKD in an AMPK/SIRT1-independent pathway in *db/db* mouse models of type 2 diabetes [24]. This recent finding appears to contradict the results of the aforementioned studies, and the exact reason for the different outcomes remains unclear (Figure 4).

##### Resveratrol Protects against Age-Related Kidney Damage by Activating SIRT1

SIRT1 expression decreases in aging kidneys, and Chuang et al. reported that reduced SIRT1 expression in podocytes aggravates glomerulosclerosis and albuminuria in aged mice and was accompanied by increased urinary 8-OH-dG levels, which is a marker of oxidative stress [201]. Kume et al. found that mitochondrial damage in aged kidneys was associated with a decrease in SIRT1 activation [140]. In the renal proximal tubular cells of aged mice, autophagy in response to renal hypoxia decreased, resulting in renal dysfunction and histological renal fibrosis. Calorie restriction-mediated renal SIRT1 activation deacetylated and activated FOXO3a transcriptional activity, leading to the recovery of BCL2/adenovirus E1B 19 kDa protein-interacting protein 3 (Bnip3)-mediated autophagy, even in aged kidneys. These findings indicated that SIRT1 is a crucial target in aging kidneys; therefore, resveratrol is expected to prevent renal aging [132]. Research is needed to evaluate the therapeutic potential of resveratrol in renal pathologies through SIRT1 modulation.

##### Piceatannol

Piceatannol (trans-3,3’,4,5’-tetrahydroxystilbene) is a naturally occurring polyphenolic stilbene found in various fruits and vegetables and has been reported to exhibit anticancer and anti-inflammatory properties. In addition, the recently reported beneficial effects of piceatannol on hypercholesterolemia, atherosclerosis, and angiogenesis underscore its therapeutic potential in cardiovascular diseases [252]. Piceatannol shows several biological activities, including anti-inflammatory, antioxidative, antiallergic, antiadipogenic, and antihypertrophic effects [252,253,254,255,256,257]. However, only a few studies have shown that piceatannol plays a beneficial role in kidney diseases. A study conducted by Fernandez et al. [258] showed that piceatannol in combination with low doses of cyclosporine A prevented kidney allograft rejection. More recently, another study showed a mild renoprotective effect of piceatannol in obese Zucker rats [259]. Choi et al. [260] investigated the effect of piceatannol on renal fibrosis and histone deacetylase (HDAC) expression in a mouse model of UUO. They found that piceatannol suppressed extracellular matrix protein deposition, including type I collagen and fibronectin, as well as connective tissue growth factor (CTGF) and α-SMA in UUO kidneys. However, the expression of EMT marker genes, such as N-cadherin and E-cadherin, was not changed in the kidneys after UUO. Considering that activation of SIRT1 by resveratrol can alleviate renal fibrosis induced by UUO in mice [25], piceatannol may have a beneficial effect on renal fibrosis. Nevertheless, a recent study [261] showed that renal fibrosis was inhibited by treatment of UUO mice with the SIRT1/2 inhibitor sirtinol or SIRT1 inhibitor EX527 [261]. These conflicting results make it difficult to explain the functionality of SIRT1 in renal fibrogenesis and raise the possibility that the antifibrotic effect of piceatannol is due to nonspecific effects (Figure 4).

### 3.2. Flavonoid Polyphenols

#### 3.2.1. Flavonols

Flavonols are the 3-hydroxy derivatives of flavanones and probably the most common subclass of flavonoids in the human diet. Flavonols are found in different types of vegetables, such as onions, spinach, broccoli, and asparagus, as well as in several types of berries, tea, and cocoa [242,262]. They are generally present at relatively low concentrations, ≈15–30 mg/kg fresh weight. The richest sources are onions (up to 1.2 g/kg fresh weight), curly kale, leeks, broccoli, and blueberries [263].

##### Quercetin

The flavonol quercetin 2-(3,4-dihydroxyphenyl)-3,5,7-trihydroxy-4H-chromen-4-one is the most common flavonoid found in vegetables, fruits, nuts, and tea and has antioxidant and anti-inflammatory properties [222,264,265]. Quercetin has shown antiproliferative, chemoprotective, and anticarcinogenic activities and plays an important role in gene expression modulation [222] (Figure 3).

##### Blood Pressure Regulation by Quercetin Mediated by SIRT1 in the Kidney

Although the regulation of blood pressure is influenced by various factors, Na^+^ reabsorption in renal tubules is an important factor regulating blood pressure because it controls extracellular fluid volume, especially in salt-sensitive hypertension [266]. In the kidney, quercetin downregulated the renal expression of epithelial Na^+^ channels (ENaCs) in hypertensive Dahl salt-sensitive rats, and this effect was associated with a reduction in systolic blood pressure [266]. ENaC is regulated by the RAS system, which maintains the body fluid level and blood pressure by modulating Na^+^ balance. Interestingly, Zhang et al. identified SIRT1 as a novel modulator of αENaC expression in cultured renal inner medullary collecting duct cells (mIMCD3 cells) [267]. The SIRT1 interacted with Dot (disruptor of telomeric silencing)-1, a histone methyltransferase of H3K79, and enhanced its methylation activity on histone H3K79 in chromatin at the αENaC promoter, thereby repressing αENaC transcription in mIMCD3 cells [267]. This study indicated that dietary quercetin supplementation may reduce elevated blood pressure caused by high salt intake by downregulating αENaC expression in the kidney (Figure 3).

##### Quercetin Alleviates Kidney Fibrosis via SIRT1

Quercetin was able to alleviate TGFβ1-induced fibrosis in renal tubular epithelial cells by suppressing miR-21 activity. Interestingly, miR-21 is associated with the incidence of renal fibrosis [268,269]. Emerging evidence implicated accelerated renal tubular epithelial cell senescence in renal fibrosis progression. Quercetin is an activator of SIRT1 deacetylase, involved in the regulation of cellular senescence. Liu et al. evaluated quercetin-induced inhibition of renal tubular epithelial cell senescence and renal fibrosis [270]. They demonstrated that quercetin effectively attenuated the senescence of rat tubular epithelial cells and renal interstitial fibrosis by enhancing mitophagy through the activation of the SIRT1-PTEN-induced kinase 1 (PINK1)/Parkin pathway in vivo and in vitro. Cellular senescence is characterized by permanent cell cycle arrest and the loss of DNA replication. Senescence can be triggered by several cellular stressors, including mitochondrial dysfunction, inflammation, DNA damage, oxidative stress, and abnormal epigenetic modification, each of which is a common characteristic pathology in CKD [270] (Figure 3).

#### 3.2.2. Isoflavones

Isoflavones belong to a group of flavonoid phytoestrogens in natural nonsteroidal phenolic plant compounds. They can be found in the Leguminosae family as nonactive hydrophilic glycosides (e.g., daidzein, genistein, and glycitein in soybean) and 4′-methylated lipophilic derivatives (e.g., formononetin and biochanin A in red clover) [271]. The main dietary sources of isoflavones in humans are soybean and soybean products, which mainly contain daidzein and genistein [272]. Phytoestrogen dietary supplements made from red clover extracts indirectly provide a source of daidzein as the methoxylated isoflavone formononetin from red clover is effectively transformed into daidzein in the human gastrointestinal tract [273] (Figure 3).

##### Genistein

Genistein (40,5,7-trihydroxyisoflavone), the soy isoflavone most extensively studied thus far, is a polyphenolic nonsteroidal compound commonly used as a dietary supplement. Because genistein exerts estrogen-like biological activity, its biological effects have been explored in conditions such as cancer, inflammation, and apoptosis [274]. Notably, genistein is closely related to SIRT1 activity [275]. Activation of SIRT1 attenuates I/R-induced renal injury; in contrast, the ablation of one allele of the SIRT1 gene significantly results in higher susceptibility to I/R-induced kidney injury. In a study conducted by Li et al., it was demonstrated that the expression of SIRT1 was significantly increased 24 h after renal I/R injury following pretreatment with genistein (10 and 15 mg/kg). Additionally, the inhibition of SIRT1 activity or expression abolished the protective effects of genistein on I/R-induced renal injury [276]. Genistein treatment significantly reduced the apoptosis rate and enhanced the cellular proliferation rate, further corroborating findings on the protective effect of genistein as mediated by SIRT1 [276] (Figure 3).

##### Formononetin

Formononetin-7-hydroxy-3(4-methoxyphenyl)chromone (C_16_H_12_O_4_) is the major compound of the *Astragalus membranaceus* and *Astragalus mongholicus* root, but it can also be found in the leaves and flowers of *Trifolium pratense*, the roots of *Glycyrrhiza glabra* and *Glycyrrhiza uralensis*, and in the root of kudzu [277]. This isoflavonoid is also evident in dietary products, such as beans, carrots, cauliflower, green peas, iceberg lettuce, and red potatoes [278]. Formononetin is known for its cardioprotective and antihyperlipidemic effects [277]. It has also been reported to reduce the generation of ROS and to have significant antioxidant potential [279]. A 16-week formononetin treatment led to improved glycemic parameters and significantly enhanced the creatinine clearance that was associated with increased SIRT1 expression in the kidney [280]. Moreover, isoflavone supplementation reduces the urinary albumin excretion level and decreases the urine albumin-to-creatinine ratio, which indicates a delayed progression to DKD in *db/db* mice (Figure 3).

##### Puerarin

Puerarin is an isoflavonoid extracted from the Chinese medical herb *Radix puerariae*. The chemical name that describes the molecular structure of this compound is 7-hydroxy-3-(4-hydroxyphenyl)-1-benzopyran-4-one-8-β-D-glucopyranoside. *R. puerariae* regulates several physiological functions, including anti-inflammatory and antioxidant processes [281]. Xu et al. evaluated the effect of puerarin on DKD in mice. Fasting blood glucose, BUN, and serum creatinine, as well as urine protein levels, were effectively ameliorated in DKD mice treated with puerarin [282]. Attenuated kidney tissue damage was observed after puerarin administration. Interestingly, an ultrastructure investigation revealed a dramatically ameliorative effect of puerarin on mitochondrial damage. In addition, the SIRT1, FOXO1, and PGC1α expression was significantly upregulated at the protein level by puerarin administration in the renal cortex (Figure 3).

## 4. Other Natural Compounds That Modulate SIRT1 Activity in the Kidney

### 4.1. Catalpol

*Rehmannia glutinosa* is a traditional Chinese herbal medicine that is thought to invigorate the kidney. *R. glutinosa* is widely used to treat a variety of ailments [283]. Catalpol, which is a biologically active ingredient of *R. glutinosa*, exhibits a variety of biological activities, including antioxidation, anti-inflammatory, antiapoptotic, and hypoglycemic effects [284]. In a study conducted by Zhang et al., the SIRT1 expression was strikingly decreased in adriamycin-damaged kidney tissue and podocytes, and catalpol treatment significantly upregulated the expression of SIRT1. Moreover, the effect of catalpol activation on SIRT1 was inhibited, and the levels of SIRT1-regulated downstream cytokines (TNFα and IL-6) increased, resulting in the aggravation of apoptosis and inflammation when the podocytes were incubated with SIRT1 siRNA or a SIRT1-specific inhibitor. These findings strongly suggested that SIRT1 plays a crucial role in the protective effect of catalpol on adriamycin-induced kidney damage [285] (Figure 3).

### 4.2. Astragaloside IV

Astragaloside IV, a bioactive saponin extracted from the *Astragalus* root, exerts a variety of pharmacological effects via its antioxidative and anti-inflammatory functions [286]. In a recent study, astragaloside IV protected against podocyte injury by enhancing autophagy in an STZ-induced diabetic mouse model [286]. Additionally, Wang et al. showed that astragaloside IV activates SIRT1, which then inhibits NF-κB p65 subunit acetylation, which culminates in enhanced autophagy in hyperglycemia-injured podocytes [287,288]. These findings suggested that astragaloside IV-mediated autophagy protects against mesangial cell injury under hyperglycemic conditions through the SIRT1/NF-κB pathway (Figure 3).

## 5. Bioavailability of Polyphenols

The proposed benefits of polyphenols, either as protective/prophylactic substances or as therapeutic molecules, is achieved mainly by the consumption of a natural polyphenol-enriched diet. It has also been shown that the health effects of polyphenols depend on the amount consumed and the component bioavailability [289]. The bioavailability of polyphenols represents a large hurdle, as they reach target organs only in very low concentrations [290]. Dietary polyphenols are predominantly present in glycosylated form with one or more sugar residues conjugated to a hydroxyl group or the aromatic ring (flavanols are one notable exception) [291,292]. This structure is the main reason for their low absorption in the stomach, and only aglycones and some glucosides can be absorbed in the small intestine, with the remainder absorbed in the colon [291]. In comparison to the intestine, the colon does not readily absorb polyphenols. This situation leads to longer absorption times, which can last for 9 h [291]. Glucosides in the food sources of polyphenols enable faster and more efficient absorption of polyphenols [291].

However, the aglycones of some isoflavones show superior absorption to their glycosylated forms [291]. Isoflavones represent the best-absorbed polyphenols, along with gallic acid, followed by catechins, flavanones, and quercetin glucosides [293]. On the other hand, proanthocyanidins, galloylated tea catechins, and anthocyanins are absorbed the least efficiently [291]. Conceptually, polyphenols are absorbed by passive diffusion [291]. Finally, polyphenol overconsumption may raise safety concerns due to polyphenol accumulation in the organism as most polyphenols undergo phase II conjugation and are rapidly eliminated in urine and bile as glucuronides and sulfate esters [294]. In kidneys compromised by any pathology, the elimination pathway of these polyphenols may be altered. This possibility is particularly important when the loose regulatory legislation regarding the commercialization and use of food supplements is considered.

## 6. Final Remarks

The study of mammalian sirtuins, particularly SIRT1, shows that numerous protein targets are involved in age-related pathogenesis and metabolism-related renal disorders, suggesting that the modulation of SIRT1 activity may exert strong renoprotective effects. Kidney damage is a result of metabolic disruption, inflammation, impaired autophagy, hypoxia, oxidative stress, apoptosis, and fibrosis, leading to diseases such as AKI, CKD, renal lipid accumulation during obesity, and DKD. The downregulation of SIRT1 deacetylase activity and its expression in renal disorders leads to the increased acetylation of histones and/or the crucial transcription factors involved in promoting the development and progression of age- and metabolism-related kidney diseases (Figure 1).

As the understanding of SIRT1 activity and NAD^+^ metabolism in renal physiology increases, many different pharmacological approaches are being developed to maintain SIRT1 activity and increase NAD^+^ production in age- and metabolism-related disorders, with the purpose of ameliorating kidney injury in animal models of renal diseases. Therapies effective in animal models need to be assessed in human clinical trials. This practice is supported by in vitro studies that have led to the identification of the molecular effects of synthetic and natural SIRT1-activating compounds in various renal cell types, demonstrating protective effects against renal disorders through modulation of SIRT1 downstream target proteins (Figure 1 and Figure 2).

Most natural SIRT1-activating compounds belong to the polyphenol family, whose members are capable of significantly increasing SIRT1 activity. Flavonoids and nonflavonoids, among other natural components, are useful in the design of more potent and specific synthetic activators that have been able to selectively enhance SIRT1 deacetylase activity and limit kidney damage. Given that SIRT1 is a crucial mediator in preventing and treating the progression of several kidney-associated diseases and possibly in improving human quality of life and extending healthspan, there is an imperative need for further basic investigations to elucidate the molecular mechanisms of the SIRT1 interactions, as well as to develop safe and effective therapeutic strategies to restore SIRT1 activity and its renoprotective and health-benefitting effects (Figure 3 and Figure 4).

## Figures and Tables

**Figure 1 nutrients-14-00653-f001:**
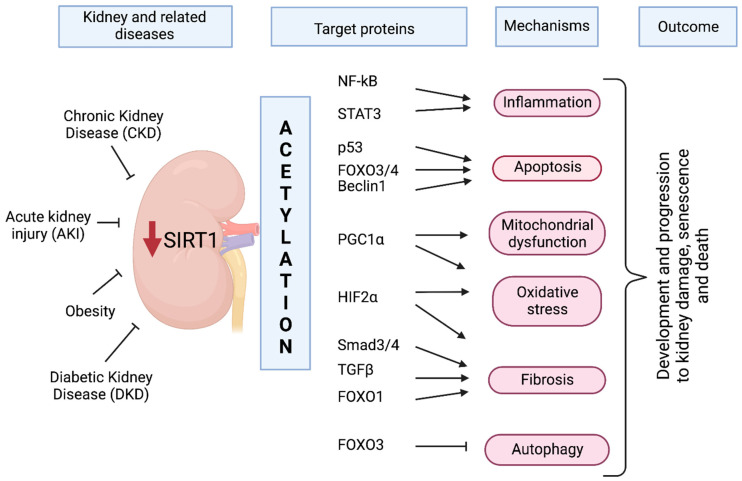
Dysregulation of SIRT1 in kidney-related diseases and the target proteins involved. SIRT1 deacetylase activity and expression are reduced (red arrow), which increases acetylation/activation of several transcription factors, leading to aggravated effects, such as inflammation, apoptosis, fibrosis, oxidative stress, and autophagy dysfunction in kidney tissue. Together, these processes contribute to the development and progression of kidney-related disorders. Therefore, the main role of SIRT1 is to protect the kidney by preserving renal homeostasis. Stimulation, ➜; inhibition, ⟞; red down arrow.

**Figure 2 nutrients-14-00653-f002:**
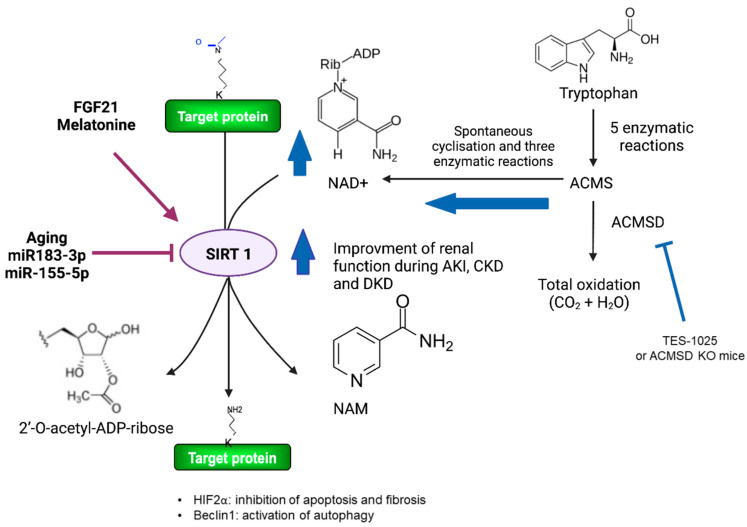
Modulation of NAD^+^ metabolism improves renal function during AKI and chronic kidney disease. SIRT1 activity is dependent on NAD^+^ levels, which are maintained, in part, through NAD^+^ biosynthetic pathways. High levels of ACMSD in the kidney decrease NAD^+^ levels. Thus, the use of ACMSD inhibitors such as TES-1025 has the potential to be efficient in protecting renal function, as observed in ACMSD-KO mice. Furthermore, SIRT1 activity is positively modulated in the kidney by FGF21 and melatonin. In contrast, aging and miRNAs, such as miR183–3p and miR155–5p, decrease SIRT1 activity. Finally, HIF2α and Beclin1 are among the newly described SIRT1 target proteins in the kidney. SIRT1, sirtuin 1; NAD^+^, nicotinamide adenine dinucleotide; NAM, Nicotinamide; ACMS, α-amino-β-carboxymuconate-ε-semialdehyde; ACMSD, α-amino-β-carboxymuconate-ε-semialdehyde decarboxylase; KO, knockout; AKI, acute kidney injury; CKD, chronic kidney disease; DKD, diabetic kidney disease.

**Figure 3 nutrients-14-00653-f003:**
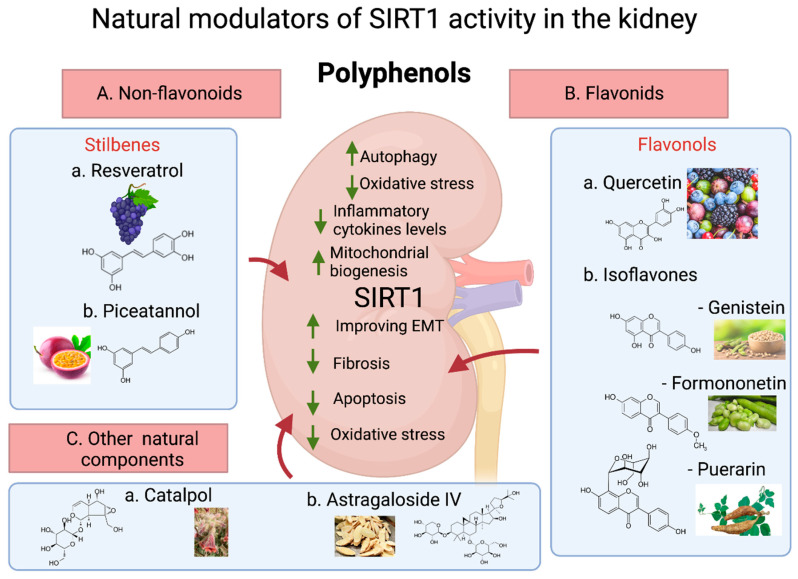
Natural modulators of SIRT1 activity in the kidney. Polyphenols are classified as (a) nonflavonoids and (b) flavonoids. SIRT1 activation by (**A**) (a) resveratrol and (b) piceatannol, nonfla-vonoid stilbenes, has shown a beneficial effect on the kidney. The flavonols (**B**) (a) quercetin and (b) the isoflavones genistein, formononetin, and puerarin have shown important beneficial effects on kidney function. Other natural components, such as (**C**) (a) catalpol and (b) astragaloside IV, have also been demonstrated to activate SIRT1 and ameliorate kidney abnormalities. Up arrow(↑), Increase mechanism; down arrow(↓), Decrease mechanism.

**Figure 4 nutrients-14-00653-f004:**
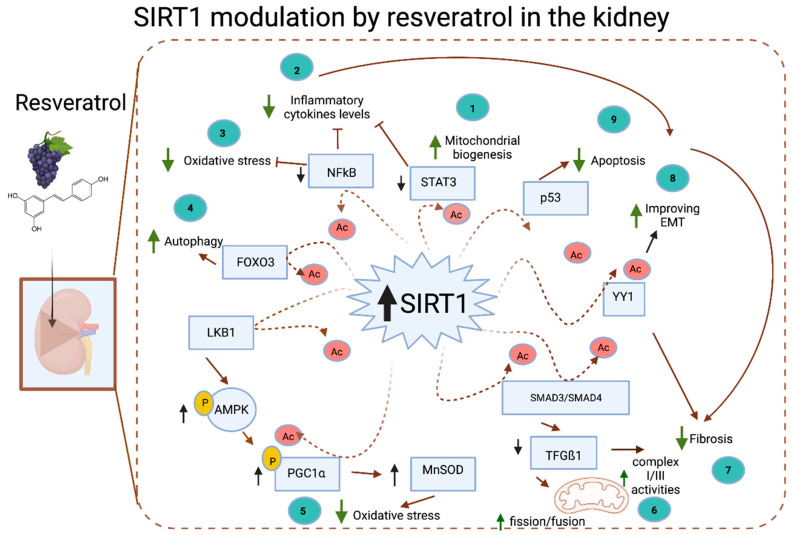
Resveratrol is found in the skin of grapes, berries, and peanuts and is a known activator of SIRT1. SIRT1 deacetylates several proteins involved in kidney homeostasis. Among these proteins, STAT3 and NF-κB are deacetylated, which promotes increased mitochondrial biogenesis (1), decreased inflammatory cytokine levels (2), and reduced oxidative stress (3). FOXO3 deacetylation increases autophagy (4); LKB deacetylation promotes AMPK phosphorylation, which results in PGC1α phosphorylation, and in conjunction with SIRT1 deacetylation, PGC1α induces a decrease in oxidative stress (5). SMAD3 and SMAD4 are deacetylated, inducing a decrease in TGFβ, which results in an increase in fission/fusion proteins and augmented mitochondrial complexes I and III (6), promoting and increasing the mitochondrial biogenesis rate; YY1 acetylation inhibits fibrosis (7) and the EMT (8); and p53 acetylation decreases the apoptosis rate (9). SIRT1, sirtuin 1; STAT, signal transducer and activator of transcription protein; NF-κB, nuclear factor kappa-light-chain-enhancer of activated B cells; FOXO, forkhead box transcription facto; LKB, liver kinase B; AMPK, 5’-adenosine monophosphate-activated protein kinase; PGC1α, peroxisome proliferator-activated receptor-gamma coactivator 1 alpha; SMAD, mothers against decapentaplegic homolog 3; TGFβ, transforming growth factor-beta; YY1, yin yang 1 protein; P with yellow, Phosphorylation; Ac, Acetylation; green arrows, The sense of the arrow indicates increased or decreased mechanism; black arrows, The sense of the arrow indicates Increased or decreased target proteins; red arrows, continuous/discontinuous.

**Table 1 nutrients-14-00653-t001:** Lysine residues deacetylated by SIRT1 in target proteins.

Target Protein	Deacetylated Lysine Residues	Function	References
NF-κB	K218, K221, and K310	Controls transcription of genes, cytokine production and cell survival; regulates innate and adaptive immune responses and carcinogenesis; is implicated in processes of synaptic plasticity and memory. K218 and K221 are involved in DNA binding. K310 participates in the transcriptional activity of NF-κB.	[31,32,33,34]
STAT3	K685, K679, K707, and K709	Mediates expression of several genes in response to cell stimuli and plays a role in cell growth and apoptosis.	[35,36,37]
FOXO1	K242, K245, K248, K262, K265, K274, K294, and K559	Regulates gluconeogenesis and glycogenolysis, functions as a tumor suppressor, and is involved in cell arrest, apoptosis, and adipogenesis.	[38,39,40,41,42,43,44]
FOXO3	K203, K242, K245, K259, K270, K271, K290, and K569	Induces cell cycle arrest, confers resistance to oxidative and genotoxic stress, functions as a tumor suppressor, and inhibits FOXO-induced apoptosis.	[41,42,44,45,46]
FOXO4	K186, K189, K215, K237, and K407	Regulates oxidative stress signaling, longevity, insulin signaling, cell cycle progression, neural differentiation, and apoptosis and functions as a tumor suppressor.	[41,44,47,48]
p53	K320, K373, and K382	Functions as a tumor suppressor and induces cell cycle arrest, senescence, apoptosis, and the response to DNA damage and oxidative stress to prompt DNA repair.	[31,49,50,51,52,53,54,55,56]
Beclin1	K430 and K437	Regulates autophagy and cell death and plays a role in tumorigenesis and neurodegeneration.	[29,57,58]
PGC1α	K77, K144, K183, K253, K270, K277, K320, K346, K412, K441, K450, K757, and K778	Functions as a transcriptional coactivator that regulates genes involved in mitochondrial biogenesis and energy metabolism.	[15,59,60,61]
HIF1α	K674	Functions as a transcriptional regulator of cell and developmental responses to hypoxia and is implicated in cancer, angiogenesis, energy metabolism, cell survival, and tumor invasion.	[62,63,64,65]
HIF2α	K741	Induces gene expression that is regulated by hypoxic stress.	[63,66,67]
SMAD3	K333, K341, K378, and K409	Mediates intracellular signaling of transforming growth factor-beta (TGF-β) and is thus related to tumor growth in cancer development.	[25,68,69]

Beclin1, Coiled-Coil, Moesin-Like BCL2-Interacting Protein; FOXO, forkhead box transcription factors; HIF, hypoxia-inducible factor; NF-κB, nuclear factor kappa-light-chain-enhancer of activated B cells; p53, tumor protein p53; PGC1α, peroxisome proliferator-activated receptor-gamma coactivator 1 alpha; SMAD, mothers against decapentaplegic homolog 3; STAT3, Signal transducer and activator of transcription 3.

## Data Availability

The study did not report any data.

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
