# Peer review of "Potential of Polyphenols to Restore SIRT1 and NAD+ Metabolism in Renal Disease"

_nutrients, 2022, doi:10.3390/nu14030653_

Round 1
Reviewer 1 Report
The paper is well done and planned. Many thoughts are given towards the significance of the sirtuin modifications. The use of drug modulators that can reduce the risk of metabolic disorders, including kidney disease against this background, is relevant.
Author Response
Reviewer 1:
The paper is well done and planned. Many thoughts are given towards the significance of the sirtuin modifications. The use of drug modulators that can reduce the risk of metabolic disorders, including kidney disease against this background, is relevant.
Response : We thank the reviewer for his/her positive comments.
Reviewer 2 Report
|
Rows chapter |
Oryginal text |
Comment |
|
119-125 |
The de novo NAD+ biosynthetic path-way is particularly relevant in the kidney since this organ expresses all NAD+ biosynthetic enzymes, such as indoleamine 2,3-dioxygenase (IDO), quinolinate phosphoribosyltransferase (QPRT), and α-amino-β-carboxymuconate-ε-semialdehyde decarboxylase (ACMSD). Notably, ACMSD forms α-aminomuconate-ε-semialdehyde (AMS), which can be oxidized to CO2 and H2O, inhibiting NAD+ synthesis and reducing NAD+ levels. Genetic and pharmacological inhibition of ACMSD increases NAD+ levels, improving mitochondrial and kidney function |
Please clarify. |
|
127 and other lines |
ACSMD |
Is it the same as ACMSD? Please provide explanations |
|
533-535 |
Glomerular damage also causes endothelial dysfunction and tubular injury, leading to cell death, endothelial-/epithelial-to- mesenchymal transitions, cell infiltration, tubule degeneration, and interstitial fibrosis mediated through different mechanisms |
Please clarify |
|
545-547 |
In glomerular visceral epithelial cells or podocytes, which are highly specialized cells in the kidney glomerulus that are essential for preventing plasma proteins from entering 546 the urinary ultrafiltrate and are key participants in the formation of the glomerular filtration barrier |
Please clarify |
|
Figure 4 |
|
Please show the resveratrol grip point |
|
3.1.1.1.4 |
.Whole chapter |
Paucity of arguments describing the role of resveratrol. |
|
861-863 |
Nevertheless, a recent study [277] showed that renal fibrosis was inhibited by treatment of UUO mice with the SIRT1/2 inhibitor sirtinol or SIRT1 inhibitor EX527 |
Please clarify |
The article is very confusing and difficult to read. Chapters should be shortened.
Combine the chapters on obesity and diabetes t.2
The content of some chapters is partially duplicated.
Reviewer 3 Report
The author wrote review article about SIRT1 and NAD in kidney disease. This article covered detail mechanism of SIRT1 and NSD from basic research to clinical evidence.
I think this article is well written and informative for nephrologist. English grammar is appropriate and easy to understand.
Author Response
Reviewer 3
The author wrote review article about SIRT1 and NAD in kidney disease. This article covered detail mechanism of SIRT1 and NSD from basic research to clinical evidence. I think this article is well written and informative for nephrologist. English grammar is appropriate and easy to understand.
We thank the reviewer for his/her positive comments.